# Position: Benchmarking is Broken - Don't Let AI be its Own Judge

**Zerui Cheng**[1,*]    **Stella Wohnig**[2,*]    **Ruchika Gupta**[3,*]    **Samiul Alam**[4,*]
**Tassallah Abdullahi**[5]    **João Alves Ribeiro**[6]    **Christian Nielsen-Garcia**[7]    **Saif Mir**[4]
**Siran Li**[8]    **Jason Orender**[9]    **Seyed Ali Bahrainian**[8]    **Daniel Kirste**[10]
**Aaron Gokaslan**[11]    **Carsten Eickhoff**[8,†]    **Ruben Wolff**[12,†]

[1] Princeton University    [2] CISPA Helmholtz Center for Information Security
[3] Michigan State University    [4] Ohio State University    [5] Brown University
[6] Massachusetts Institute of Technology    [7] University of California, Los Angeles
[8] University of Tübingen    [9] Old Dominion University    [10] Technical University of Munich
[11] Cornell University    [12] Forest AI
[*] Equal Contributions. [†] Advisors.

`zerui.cheng@princeton.edu`

## Abstract

The meteoric rise of Artificial Intelligence (AI), with its rapidly expanding market capitalization, presents both transformative opportunities and critical challenges. Chief among these is the urgent need for a new, unified paradigm for trustworthy evaluation, as current benchmarks increasingly reveal critical vulnerabilities. Issues like data contamination and selective reporting by model developers fuel hype, while inadequate data quality control can lead to biased evaluations that, even if unintentionally, may favor specific approaches. As a flood of participants enters the AI space, this "Wild West" of assessment makes distinguishing genuine progress from exaggerated claims exceptionally difficult. Such ambiguity blurs scientific signals and erodes public confidence, much as unchecked claims would destabilize financial markets reliant on credible oversight from agencies like Moody's.

In high-stakes human examinations (e.g., SAT, GRE), substantial effort is devoted to ensuring fairness and credibility; why settle for less in evaluating AI, especially given its profound societal impact? **This position paper argues that a laissez-faire approach is untenable. For true and sustainable AI advancement, we call for a paradigm shift to a unified, live, and quality-controlled benchmarking framework—robust by construction rather than reliant on courtesy or goodwill.** Accordingly, we dissect the systemic flaws undermining today's evaluation ecosystem and distill the essential requirements for next-generation assessments.

To concretize this position, we introduce the idea of PeerBench, a community-governed, proctored evaluation blueprint that seeks to improve security and credibility through sealed execution, item banking with rolling renewal, and delayed transparency. PeerBench is presented as a complementary, certificate-grade layer alongside open benchmarks, not a replacement. We discuss trade-offs and limits and call for further research on mechanism design, governance, and reliability guarantees. Our goal is to lay the groundwork for evaluations that restore integrity and deliver genuinely trustworthy measures of AI progress.

39th Conference on Neural Information Processing Systems (NeurIPS 2025) Position Paper Track.

# 1 Introduction

The widespread adoption of AI technologies — especially foundation models (FMs) — in decision-making processes has considerably heightened their societal impact. As a result, the need for the rigorous assessment of their performance has become increasingly urgent, positioning AI evaluation as a critical area of study. Benchmarks have become such influential forces in the AI industry that companies reportedly invest hundreds of thousands of dollars in compute resources to achieve top scores on evaluations such as the ARC-AGI benchmark [18]. Following the work of [20], we define a benchmark as a specific pairing of one or more datasets (typically including a test set, and sometimes training data as well) and an evaluation metric. This combination is intended to represent a particular task or set of capabilities, and is adopted by a research community as a common framework for comparing different methods. Benchmark leaderboards have become the go-to standard for evaluating the progress across AI subfields -from ImageNet[22] in vision to GLUE[23] in language. Evaluating algorithmic progress with benchmarks has become a double-edged sword; while they have accelerated iteration and competition, their popularity has also incentivized chasing of state-of-the-art (SOTA) performance [21], making them vulnerable to overfitting, gaming, and selective reporting. For instance, models that achieve so-called "superhuman" performance on question answering leaderboards often fail dramatically when evaluated on out-of-distribution inputs, revealing a lack of true understanding.

In order to better understand what benchmarks are truly measuring, The Markup [24], an investigative newsroom under CalMatters, interviewed researchers who designed evaluation datasets which revealed that many widely-used benchmarks are years old, increasing the likelihood that they were included in training data—compromising their effectiveness as unbiased evaluation tools. Public datasets often find their way, intentionally or inadvertently, into the training corpora of large models [25, 27, 28, 29], enabling memorization of test items rather than true generalization. Benchmark designers themselves may, intentionally or not, cherry-pick examples that favor particular architectures under pressure to produce impressive results. On the other end of the spectrum, proprietary or pay-walled evaluations limit accessibility and rely on the continued goodwill of their owners to remain relevant. Collectively, these practices create a distorted landscape in which leaderboard positions can be manufactured, scientific signal is drowned out by noise, and community trust is eroded[30].

Many scholars have raised concerns about the limitations of AI benchmarking, with some describing current evaluation practices as a "minefield" [26]. As hype increasingly overshadows genuine progress, the need for rigorous, trustworthy evaluation has become critical, especially when introducing new paradigms. In the following paragraph, we examine key structural flaws in the current evaluation pipeline, including data contamination, fragmented and inconsistent benchmarks, opaque dataset curation, the lack of safeguards for fairness and freshness, and how these issues have enabled superficial progress while undermining trust across the AI community.

**Cracks in the Current Paradigm.**

- **Data Contamination.** Public benchmarks may leak into or be deliberately injected into training sets, leading to test-set memorization and inflated scores [31, 32]. With today's large-scale models trained on multi-trillion-token corpora, such contamination is increasingly inevitable [46, 51]. Retrieval-based audits report over 45% overlap on QA benchmarks, and GPT-4 infers masked MMLU answers in 57% of cases—well above chance [44]. Allegations that LLAMA 4 gained significant improvements via seeded paraphrases illustrate how easily scores can be engineered. N-gram audits, like those used on Qwen-1_8B [45], can help detect leakage but rely on partial knowledge of training data. Once contamination is plausible, generalization claims become suspect [37, 38, 35].

- **Risk of Strategic Cherry-picking.**

  - **Collusion.** Benchmark creators may collude with model creators and create hand-crafted suites that inadvertently or strategically advantage particular AI models.

  - **Selective Reporting.** Model creators can highlight performance on favorable task subsets, creating an illusion of across-the-board prowess, and preventing the audience from having a comprehensive bird's eye view of the current landscape.

- **Bias in Test Data.** Current benchmarks, lacking unified data quality control, frequently suffer from test data bias, which can be an intentional or unintentional outcome of their design. This can lead to fundamentally misleading evaluations. For example, in Humanity's Last Exam [1], organizers select five specific models and curate tests consisting solely of items that all five chosen models fail. Performance scores on such a dataset would unfairly penalize the initial five models and create an artificial advantage for any new model. This is akin to evaluating two models, A and B, of equal intrinsic ability on a task distribution $\mathcal{D}$ (both solving 50% of tasks). If the test set is then constructed using only the subset of $\mathcal{D}$ that model A solves but model B fails, it generates a specious conclusion of A's superior performance, obscuring the fact that the test data itself is unrepresentative and biased.

- **Dataset Collection.** One key structural issue is the devaluation of dataset work within the machine learning community. In contrast to model innovation, dataset curation and documentation are treated as lower-status contributions. This has led to a culture in which datasets are frequently "reduced, reused, and recycled" without thorough contextualization [33], complicating efforts to track biases. Park and Jeoung [34] further observe that benchmark-sharing platforms like PapersWithCode suffer from inconsistent metadata terminology. Key details such as licensing and annotation processes are often missing, which complicates standardization efforts.

- **Noisy metrics, hypes & Evaluation fragmentation.** Public benchmark suites suffer from severe heterogeneity—each repository often introduces custom tokenizers, scoring rules (e.g., BLEU, ROUGE, EM, proprietary AI-scores), and ad-hoc scripts [49], making results difficult to reproduce and compare. Nearly all benchmarks are *static*, with performance gains increasingly reflecting task memorization rather than capability. For example, SUPERGLUE was rapidly saturated, with LLMs hitting performance ceilings shortly after release [53]. The lack of *liveness*—continuous inclusion of fresh, unpublished items—renders today's metrics a stale snapshot. These inconsistencies encourage hype-driven "state-of-the-art" claims [36], misguide resource allocation, and crowd out rigorous analysis. Recent work [48, 50] calls for standardized, live evaluation protocols to reduce overhead, unify benchmarking efforts, and establish a shared understanding of *what to beat, how to measure it, and where the true frontier lies*.

- **Restricted accessibility for Private Benchmarks.** Proprietary or paywalled benchmarks can reduce contamination [56], but they shift epistemic authority to the curator, who alone controls evaluation access, task updates, and scoring [55]. This centralization raises ethical concerns: scientific progress becomes contingent on opaque processes, discretionary labor, and sustained funding. Without transparency in item selection, bias control, and submission filtering, it is unclear whether reported gains reflect true capability[39, 35] or favorable curation. Meanwhile, benchmark legitimacy is often conferred through peer review or citation momentum rather than principled design [39, 40]. As history shows, once interest fades, such benchmarks stagnate, yet continue to shape perception and citations [41].

- **Lack of Fairness and Proctoring.** Unlike high-stakes human exams, AI evaluations lack proctors, identity checks, and appeals processes [47]. Teams may fine-tune on test sets, exploit unlimited submissions, or selectively report results, often within current norms. Cultural, linguistic, and demographic skews [52] further bias outcomes, yet no oversight body governs these axes. This creates an uneven playing field, where resource-rich teams can game the system while more cautious researchers underreport.

Together, these factors blur scientific signals and undermine confidence in reported progress.

These shortcomings are particularly stark when compared to standardized human assessments like the SAT, GRE, or bar examinations, which are proctored, regularly updated, and governed by rigorous procedures to uphold fairness, reliability, and data integrity [43, 54]. *Why do we hold machines to lower evaluative standards than we do for humans in high-stakes environments?*

**Desiderata for Next-Generation Evaluation.** An ideal modern benchmarking regime should therefore be:

- **Unified**. All benchmarks operate under a single governance framework with common interfaces, standardized result formats, and a shared execution environment. A leaderboard,

akin to a "HuggingFace for evaluation", lets researchers see at a glance where every model stands and removes the friction of juggling incompatible test harnesses.

- **Comprehensive**. The suite spans every major modality and task family, from individual modalities to multimodal reasoning, so progress can be tracked holistically rather than in isolated silos. Developers immediately know which capability gaps remain and which benchmarks to target next, without trawling the Internet for niche datasets.

- **Live and consistent**. Fresh, unpublished tests are produced on a rolling basis, preventing overfitting and test-set memorization, while earlier tests are retired and made public for auditing and research purposes. Robust score-normalization procedures align results across cohorts, ensuring that models evaluated on different slices of the benchmark remain directly comparable over time. To further preserve temporal validity, score decay methodologies, such as logistic time decay, can be applied to discount stale results and reflect the evolving relevance of model capabilities as both training regimes and real-world usage contexts shift.

- **Quality-controlled**. Each test, after being made public, is peer-reviewed for originality, difficulty, and bias, and its influence on a model's composite score should be weighted by a transparent reputation system. This mechanism is crucial to down-weight low-quality or adversarial items, deter collusion between test authors and model developers, and preserve the integrity of the signal.

Any viable successor must deliver **contamination resistant, metric unification, transparent yet decentralized governance, and auditable fairness guarantees** — principles that define the vision for next-generation AI benchmarking.

We introduce a prototype of the desired paradigm, PEERBENCH, a community-powered platform for AI evaluation that demonstrates the practicality and outlines a roadmap toward this ultimate goal.

**In summary, we posit that AI benchmarking paradigm should be reimagined and unified for built-in trustworthiness, data quality control, and contamination immunity**. Unlike traditional benchmarks-static artifacts designed and maintained by closed teams—PEERBENCH proposes a shift toward evaluation as a living, auditable process governed by transparent rules and fueled by ongoing validator contributions, evolving with the field. On top of that, we design a prototype of the desired paradigm, PEERBENCH, a community-powered platform for AI evaluation, showing the practicality and roadmap to achieving the ultimate goal.

**Key contributions.** The main contributions of our paper are:

- **Structural critique.** A formal critique of the structural flaws, contamination, fragmentation, and monopolization undermining today's benchmarks.

- **Position statement.** A position statement that reframes AI evaluation as a secure, standardized examination, together with design principles that balance openness and rigor.

- **Prototype architecture.** The PEERBENCH design is a minimum viable version of the desiderata, featuring a concrete ten-step workflow, cryptographically signed artifacts, a lightweight reputation scheme, and score-normalization methods that together transform heterogeneous community inputs into a longitudinal, contamination-resistant leaderboard.

The remainder of the paper is organized as follows. Section 2 reviews prior work on AI evaluation, draws lessons from human standardized testing, and systematically critiques the structural flaws of the current benchmarking regime. Building on this critique, Section 3 articulates our stance and distills the essential requirements for a next-generation evaluation paradigm. Section 4 presents PEERBENCH, a minimum-viable prototype that operationalizes these requirements through a live reputation system and liveness guarantees. Section 5 explores alternative designs, discusses current limitations, and outlines directions for future work. Finally, Section 6 concludes the paper.

## 2 Related Work

Public leaderboards have made undeniable contributions by spurring significant breakthroughs in AI; yet, the following review of current benchmarking efforts reveals persistent challenges in achieving sufficient robustness against contamination, ensuring long-term sustainability, and fostering genuine inclusiveness.

**Static Benchmarks and Leaderboards.**   Widely-used suites such as MMLU [2], GSM8K [3], and SuperGLUE deliver clear snapshots of progress, but each ships a single public test set that quickly saturates and leaks into training corpora. BIG-Bench's one-off community release [5] broadened task coverage, yet those tasks likewise became public upon publication, sharply reducing their discriminative power. HELM [6] added multiple metrics and periodic reports, but remains curator-driven and static between releases. In short, static benchmarks age poorly and cannot prevent data contamination.

**Dynamic or Contamination-Resistant Benchmarks.**   LiveBench [7] refreshes tasks continuously, demonstrating that rolling updates slow leakage. Still, it relies on a single centralized team, limiting scale and diversity, and highly depends on the creators' goodwill to actively maintain. Similarly, Dynabench [8] explored adversarial data collection with humans-in-the-loop, but its reach was limited by centralized infrastructure and annotation scalability. Adversarial "break-the-model" contests [12] expose weaknesses but run sporadically and lack systematic score aggregation. Robustness probes like Checklist [9] explore model failures via templated behavioral tests, but require hand-crafting and do not scale to sustained, community-wide evaluation. PEERBENCH extends these ideas by democratizing task sourcing and embedding adversarial challenges into a permanent, governance-backed workflow.

**Human-Preference and Open Evaluation Platforms.**   Crowdsourced pairwise ratings power Chatbot Arena's Elo ladder [10] and OpenAI Evals, while the HuggingFace Open LLM Leaderboard lets users upload test scripts. These platforms foster openness, yet they rest on static prompt sets, absence of identity verification, or vendor-specific ecosystems, making them vulnerable to spam, bot voting, and untracked contamination. The evaluation results become less convincing because of ineffective data quality control. PEERBENCH addresses those gaps with verified validators, reputation-weighted scoring, and single-use test suites.

**Reputation Systems and Decentralized Governance.**   Mechanisms from Stack Exchange, Wikipedia, and blockchain governance inspire our validator-reputation design, but no prior AI benchmark fully unifies decentralized contribution, reputation weighting, and contamination safeguards. PEERBENCH is, to our knowledge, the first to weave these strands into an end-to-end, self-sustaining evaluation network.

**Summary.** Prior work offers valuable ingredients, like broad task coverage, rolling updates, and human preference ratings, but each leaves critical weaknesses: public data leakage, single-team bottlenecks, or unverifiable crowdsourced inputs. In this position paper, we call for a new benchmarking paradigm that synthesizes these ideas while eliminating their shortcomings by design, filling a long-standing gap in trustworthy, contamination-immune evaluation.

## 3   Towards a New Paradigm: From Static Leaderboards to Proctored Exams

To remedy the failure modes catalogued in Section 2, we propose recasting AI benchmarking as a **standardized, proctored examination** rather than an "open-book" contest of self-reported scores. The analogy is deliberate: human aptitude tests (e.g. SAT, GRE, bar exams) have evolved over decades to balance security, fairness, and public credibility—precisely the properties modern AI evaluation lacks. Our paradigm rests on seven principles.

- **Secret test sets.** Evaluation items remain undisclosed until runtime. The question bank is either freshly generated or drawn from an encrypted reserve, precluding training-time contamination and rote memorization.

- **Proctored execution.** Models are evaluated in a unified sealed sandbox with an identical execution environment. The procedure mirrors a human exam: a fixed knowledge state is tested under identical, monitored conditions. All inputs and outputs are logged and cryptographically signed via hashing to prevent tampering.

- **Community governance.** A multi-stakeholder network of validators enforces rules and governance. The validator network curates test content and peer reviews test submissions. Validator actions are incentivized and audited via a transparent reputation and slashing system.

- **Continuous renewal and liveness.** At every evaluation round, a fixed fraction of questions is retired and replaced. Retired items may be released for research, but they are never reused for new score submissions once they are made public.

- **Auditability and integrity.** Validators pre-commit to test and answer hashes before publication. Later, a randomly selected public subset is revealed to allow other validators to cross-verify fidelity, prior exposure, and integrity. Proven data leakage results in disqualification, analogous to academic dishonesty.

- **Equitable access.** Any bona-fide team, academic, corporate, or independent, can submit a model, subject only to compute reimbursement fees. A small laboratory competes on precisely the same footing as a large vendor.

- **Multi-metric reporting.** Following educational testing practice, the score report provides domain-specific subscores (e.g., maths, coding, reasoning) and percentile ranks, not merely a single headline number. Fairness metrics (bias, robustness) are computed uniformly across models.

These principles demand greater up-front effort, drafting high-quality secret items, operating a proctoring infrastructure, but they yield durable benefits: contamination immunity, reproducible fairness, and results that stakeholders can audit rather than trust on faith. Table 1 surmises the contrasts with the status quo.

Table 1: Comparison of AI evaluation platforms. A desired paradigm should combine the strengths of prior approaches (fresh unseen tasks, expert involvement, human feedback, data quality control) with auditability, to mitigate weaknesses (central trust, static data, etc.).

| Benchmark | Dynamic Update | Data Source Diversity | Transparency | Contamination Resistance | Data Quality Control |
|---|---|---|---|---|---|
| *Static Evals (MMLU, etc.)* | No | Single (Originating Team) | Yes (Public test sets) | No[‡] | Opaque; Community-Reliant[†] |
| Scale **SEAL** (2023) | Yes (Continuous) | Single (Scale AI) | No (Private test sets) | Yes (Proprietary) | Opaque; Vendor-Internal[†] |
| **LiveBench** (2024) | Yes (Monthly) | Single (Research Team) | Yes (Public post-evaluation) | Partial[‡] | Opaque; Team-Internal[†] |
| **ARC-AGI** (2019–) | Partial (Episodic Sets) | Single (Organizers) | Yes (Public test sets) | Partial[‡] | Opaque; Expert-Driven[†] |
| **Chatbot Arena** (2023) | Yes (Ongoing Prompts) | Yes (Crowdsourced) | Yes (Public prompts) | N/A[§] | Limited (Elo-Based)[†] |
| **Desired Paradigm** | Yes (Continuous Rolling) | Yes (Validator Network) | Yes (Public post-evaluation) | Yes (By Design) | Transparent; Unified[*] |

[†]Refers to data quality control that is not explicitly defined by, transparent to, or verifiable by the broader community, often relying on the originating entity's internal standards or reputation.

[‡]Susceptible to contamination once test items are released or become predictable, even if new items are added.

[§]Focuses on preference-based chat quality with dynamic inputs, not fixed knowledge test sets that could leak.

[*]Achieved via explicit, community-vetted standards, reputation systems, and transparent post-hoc auditing of test items and processes.

*Note: A core challenge for achieving such a desired paradigm is **temporal fairness**.*

- If evaluation does not occur simultaneously on all evaluated models, then information can leak across models and periods (especially for models with the same creator but evaluated at different time) which induces contamination.

- If test data are created after a model appears, then contributors can cherry pick items to favor or handicap that particular model, though possibly reputation systems, carefully designed mechanisms, and cross validation may help mitigate the risk.

As a result, the necessary conditions for the fairest score to arise include

- (1) Tests are created before a model is released and remain fully secret until evaluation;
- (2) All eligible models are evaluated at the same time on the same undisclosed items, and the items should be discarded for future evaluation once used for evaluation.

This ideal is extremely demanding for test creators to continuously deliver high-quality test cases, and it reduces direct comparison between models that do not appear in the same cohort.

The remainder of the paper instantiates some of these principles in a concrete, practical, and community-governed prototype, PEERBENCH, whose architecture and methodological safeguards are presented in Section 4, demonstrating the practicality of our position. PEERBENCH prioritizes instant evaluation for new models over ultimate fairness, and reputation systems and slashing mechanisms are deployed for mitigating possible contamination risks.

## 4 PeerBench: A Live, Community Governed Benchmarking Platform

We introduce PEERBENCH, a platform that instantiates a proctored exam paradigm for model evaluation. The system combines a lightweight layer of community governance with a cryptographically

verifiable workflow. The goal is sustained data quality for tests and a live evaluation that resists contamination and remains reproducible.

## 4.1 Actors

PEERBENCH supports multiple evaluation streams such as mathematics, code generation, and translation. Participants interact with a common coordination service and a live reservoir of tests.

**Data contributors** author private test suites together with executable scoring functions. They query registered models with their tests and record responses. Each contributor maintains a reputation that evolves through peer review of their tests.

**Reviewers** evaluate the quality of submitted tests. They produce ordinal ratings that determine test weights. Reviewer reputation is the Pearson correlation between their ratings and the final consensus quality of the same tests.

**Model creators** expose inference endpoints for their models and register for specific streams. Each model is evaluated exactly once on each test before that test retires and becomes public.

**Coordination server** authenticates uploads, manages the live reservoir, schedules peer review, updates reputations, cross validates scores against external benchmarks, and publishes public leaderboards. It stores all artifacts in a database and immutable object storage for audit.

**End users** are researchers, journalists, regulators, and practitioners who consult PEERBENCH for live leaderboards. They may apply trust thresholds that reflect their tolerance for uncertainty.

## 4.2 Three Leaderboards

The platform maintains three leaderboards that update continuously.

1. **Data contributor leaderboard** ranks contributors by cumulative test quality and verification bonuses

$$\text{ContributorScore}(c) \ = \ \sum_i \text{quality}(T_i^{(c)}) \ + \ \text{bonuses}. \tag{1}$$

   This rule rewards both quality and steady contributions. Bonuses come from successful verification of track record of expertise (e.g. edu email, Google scholar profile, GitHub profile, etc.).

2. **Reviewer leaderboard** ranks reviewers by accuracy relative to consensus quality score

$$\text{ReviewerScore}(r) \ = \ \text{Pearson}\Big(\{q_r^{(i)}\}, \{\bar{q}^{(i)}\}\Big). \tag{2}$$

   This rule rewards alignment with consensus across streams, and reviewers with higher reputations will be given priorities in reviewing new prompts, while malicious reviewers identified via consistently low reputation will be punished or removed from the platform.

3. **Model leaderboard** ranks models by a weighted average that respects test quality

$$\text{ModelScore}(m) \ = \ \frac{\sum_i w(T_i)\, s_i^{(m)}}{\sum_i w(T_i)}. \tag{3}$$

   This rule takes data quality into the evaluation metric, striving to give a fair, robust, and contamination-free evaluation of the true abilities of different AI models.

## 4.3 Temporal Fairness Dilemma and Scheduling in PEERBENCH

As discussed in the previous section, a core challenge in PEERBENCH is temporal fairness. Models submitted at different times face different evaluation conditions as the test reservoir evolves, creating a fundamental tension between immediate evaluation and synchronized fairness.

**Design Choice A: Immediate Scoring on Request**  Under this paradigm, a model is scored immediately upon request. This approach maximizes responsiveness and enables continuous iteration, aligning with the widely adopted on-demand benchmarking paradigm. The system adapts naturally to fluctuating data contribution rates by maintaining only the most recent and highest-quality tests.

The reputation system and weighted scoring help mitigate contamination effects through post-hoc identification of malicious actors. However, this approach still risks contamination from earlier interactions with similar items that may have leaked into training corpora, particularly for models from the same creator, and it cannot be fully and immediately addressed through reputation. Additionally, models evaluated at different times face different test sets, complicating direct comparison.

**Design Choice B: Registration with Periodic Synchronized Evaluation**   Models register for predetermined evaluation windows following a geometric progression (e.g., $2^0, 2^1, 2^2, \ldots, 2^k$ days). At each window's end, all registered models are evaluated simultaneously on the same finalized secret test set. This guarantees the strongest form of fairness through secrecy and simultaneity, ensuring clear comparability within cohorts. However, it reduces inter-cohort comparability and increases operational burden on test creators. When data contribution fluctuates, platform liveness cannot be guaranteed as used tests are immediately made public and cannot be reused. Model creators must also wait for evaluation windows, potentially slowing development cycles.

**Platform Stance**   PEERBENCH adopts a hybrid approach, aiming to support both paradigms with a portion of data dedicated to immediate scoring and the remainder for synchronized evaluation. In the prototype, we prioritize immediate scoring for flexibility and timely feedback (i.e. Design Choice A). Synchronized windows produce gold standard cohort results, and cross-validation between both approaches provides additional confidence metrics. The coordination server records evaluation mode and cohort identifiers for every score, with public leaderboards displaying this metadata to enable informed interpretation of comparability. This hybrid strategy balances fairness, liveness, and practical utility while maintaining transparency about inherent trade-offs.

### 4.4   End-to-End Workflow of PEERBENCHPrototype

Figure 1 in Appendix A presents the overall architecture and control flow. The design uses three visible lanes for contributors, the coordination server with the live reservoir, and reviewers. Models appear as external endpoints that receive queries and return responses through authenticated calls.

#### 4.4.1   Setup Phase

Data contributors, reviewers, and model creators register with verifiable credentials such as institutional email. Each participant generates a public signing key. Contributors and reviewers stake collateral that can be slashed when misconduct is detected. Contributors receive initial reputation from verification bonuses. Reviewers begin with a neutral reputation.

#### 4.4.2   Continuous Evaluation

The system maintains a live reservoir of size $k$. Only retired tests become public. A public test never influences evaluations of models that arrive after its publication.

**T1. Test submission and commitment**  A contributor $c$ submits a test $T^{(c)}$ with an executable scoring function $F^{(c)}$. The system records a binding commitment $h = \mathrm{Com}(T^{(c)}, F^{(c)})$ that prevents retroactive cherry picking and fixes the evaluation stream.

**T2. Model evaluation**  The server schedules immediate queries to all currently registered models. The contributor stores audit logs and raw answers $A^{(c,m)}$ and computes objective scores $s^{(c,m)}$ using the provided scoring function. Each model is evaluated once on each live test. Evaluation must complete before retirement and publication.

**T3. Review process**  The server assigns the test to reviewers at random with a requirement of at least three valid reviews. Reviewers assign integer quality scores $q \in \{-1, 0, 1, 2\}$. Received reviews are visible to other reviewers for the same test. The final test quality $\overline{q}^{(i)}$ is the average of the collected scores weighted by the reputation of the corresponding reviewers.

**T4. Weight calculation**  The final weight balances test quality and contributor reputation[1]

$$w\big(T^{(c)}\big) \;=\; \max\Big\{0,\; 0.7 * \mathrm{quality}\big(T^{(c)}\big) + 0.3 * \min\big(2, \rho_c/100\big)\Big\}. \tag{4}$$

---

[1]It's an instance of the weight calculation formula, and can be subject to further discussion and change.

**T5. Reservoir management** The new test joins the live reservoir. The server retires tests at every step. Priority goes to tests with zero weight. If none exist the server retires the oldest test. Retirement triggers publication of the full artifact. This includes the test data, the score logs, the peer reviews, and the model responses. Retired tests never contribute to future evaluations.

**T6. Reputation updates** After each round, the reputation of all associated data contributors, reviewers, and models is updated according to Equations 1 2 3 respectively.

**M1. New model integration** When a new model is submitted to the platform, it is evaluated on the current reservoir for a preliminary initial score. However, the score may be affected by data contamination (e.g., exactly the same prompt may be leaked into the training corpora through earlier evaluation of another model of the same creator); and the score will eventually converge to a fair one once all tests that the model faced before its arrival are retired and published.

## 4.5 Security and Audit

**Partial revelation for online peer review** The system reveals small random portions of live tests to reviewers in a read-only and non-copyable format such as images. Reviewers verify consistency with the commitment $h$, and re-run scoring to confirm that published scores match the logs. Significant mismatch will trigger investigation and possible punishment. Finally, reviewers will assign a score based on the quality of the test (e.g., soundness, clarity, novelty, etc.).

**Full publication** After retirement the platform publishes the test $T_i^{(c)}$, the logs, and the model responses $A_i^{(c,m)}$. Anyone on the platform can check consistency with the hash commitments and flag any invalid prompts or possible misbehaviors. The central coordination server can take action on any misbehavior that successfully slips through the initial peer review but eventually gets caught by the community members after full publication.

**Slashing mechanism and Economic model** Participants whose reputation falls below a threshold are removed. In an ideal workflow, participants stake a certain amount of collateral before joining the platform, and receive rewards for positive contribution (reflected by the reputation scores on data contribution and reviewing). The platform slashes collateral for malicious tests or systematic deviation from consensus and uses the slashed collateral (and possible revenue if benchmarking on PEERBENCH platform is made as a paid service to model creators) to reward honest contributors of the platform, creating a self-sustainable economic loop.

## 4.6 How Our Design Choices Address Common Issues

- **Data contamination and cherry picking.** Validators pre-commit to test sets, which remain private until the round concludes and are never reused. This ensures that training on this data is infeasible if the validator behaves honestly. Any collusion or cherry-picking, such as tailoring questions to favored models, can be detected via cross-validator score discrepancies, discouraging misconduct through slashing and reputation penalties. This incentivizes honest behaviour.

- **Cheating on private data.** A public source of randomness determines which queries are revealed, preventing validators from anticipating which items will be audited. Hash commitments to all related artifacts ensure verifiable consistency and enable reliable detection of manipulation.

- **Test quality.** Each test receives multiple independent reviews. Data quality decides its weight in the final score of models. Low quality or biased content is discounted or removed, and continuously spamming the platform with low quality data may be subject to monetary punishment and slashing.

- **Accessibility and continuous operation.** Registration is light for all roles which supports broad participation. Continuous operation maintains evaluation momentum when participation fluctuates. Reviewers require appropriate incentives for consistent participation, with monetary compensation or platform credits as plausible options that align with the reputation system.

# 5 Alternative Views

The proposed transition from open, self-reported leaderboards to a proctored, community-governed examination system signifies a substantial evolution from current AI benchmarking practices. This

section addresses potential counterarguments and perceived limitations of our proposed paradigm, outlining the compromises and safeguards we envision to ensure its effectiveness and integrity.

### 1. Preserving the Value of Open Benchmarks

*Concern.* Public datasets fuel rapid AI progress by allowing universal access for error diagnosis and quick iteration. A shift to hidden tests could impede this vital open development cycle.

*Our Approach.* We advocate for a two-tiered system. "Practice" sets—comprising retired questions or legacy benchmarks—would remain openly accessible for ongoing debugging and method development, while a "final" set of fresh, unseen questions would determine certified scores, akin to the public/hidden data split in Kaggle competitions.

### 2. Ensuring Transparency and Trust with Secret Tests

*Concern.* If evaluation questions are kept secret, how can the research community be confident that they are free from bias towards specific methodologies or approaches?

*Safeguards.* We propose several measures: (i) The exam board will be multi-institutional with rotating membership to ensure impartiality; (ii) statistical summaries detailing topic distribution, difficulty calibration, and demographic coverage will be published with each test release; and (iii) all test items will be released after retirement, enabling thorough post-hoc scrutiny and community auditing.

### 3. Addressing Practical Costs and Logistical Hurdles

*Concern.* The resources required for developing secure questions, operating a dedicated evaluation server, and covering inference computation costs are non-trivial.

*A Feasible Path Forward.* Existing neutral organizations (such as NIST or MLCommons) or a newly established not-for-profit foundation could undertake hosting the evaluation service. Costs could be managed through a combination of modest submission fees and public funding to support academic participation. Containerized inference submissions can also be implemented to protect proprietary model weights while still allowing for secure, remote execution.

### 4. Balancing Innovation Pace and Open-Ended Exploration

*Concern.* The dynamic of instant feedback on public leaderboards often ignites creative "leaderboard hacking," which can subsequently evolve into genuine research advancements. A slower examination process might inadvertently dampen this innovative energy.

*Our Perspective.* Researchers will remain free to experiment and iterate using open data sources; the proposed exam system is designed to provide a high-confidence certificate. In practice, the inherent uncertainty regarding the exam's precise content is likely to encourage broader, more generalizable research rather than narrow overfitting. While a slower feedback loop is an acknowledged trade-off, it is justified by the significant gains in the reliability and robustness of the evaluation outcomes.

In summary, our proposed paradigm does not argue against the principle of openness in AI research but rather targets the vulnerabilities associated with *over-exposed* test sets. By integrating public "practice" data with a system of rolling, audited secret exams, we aim to uphold the collaborative spirit of the AI community while simultaneously restoring confidence in headline performance claims.

## 6   Conclusion

Benchmarking is the heartbeat of empirical AI, yet static public datasets now leak, self-reported leaderboards are gamed, and headline scores may no longer signal real ability. Inspired by human exams, we advocate replacing open-book, developer-run benchmarks with a proctored, community-governed test. The core requirements—secret tests, liveness, data quality—coalesce in our PEERBENCH design.

> **Call to Action**
>
> Progress in AI must be *measured*, not merely marketed. We invite researchers, practitioners, and policymakers to help refine, deploy, and steward this emerging evaluation paradigm. By directing collective effort toward *how* we measure, we protect the integrity of *what* we build, so that claims of "state-of-the-art" performance once again carry demonstrable scientific weight.

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

# A    Workflow of PEERBENCH Prototype

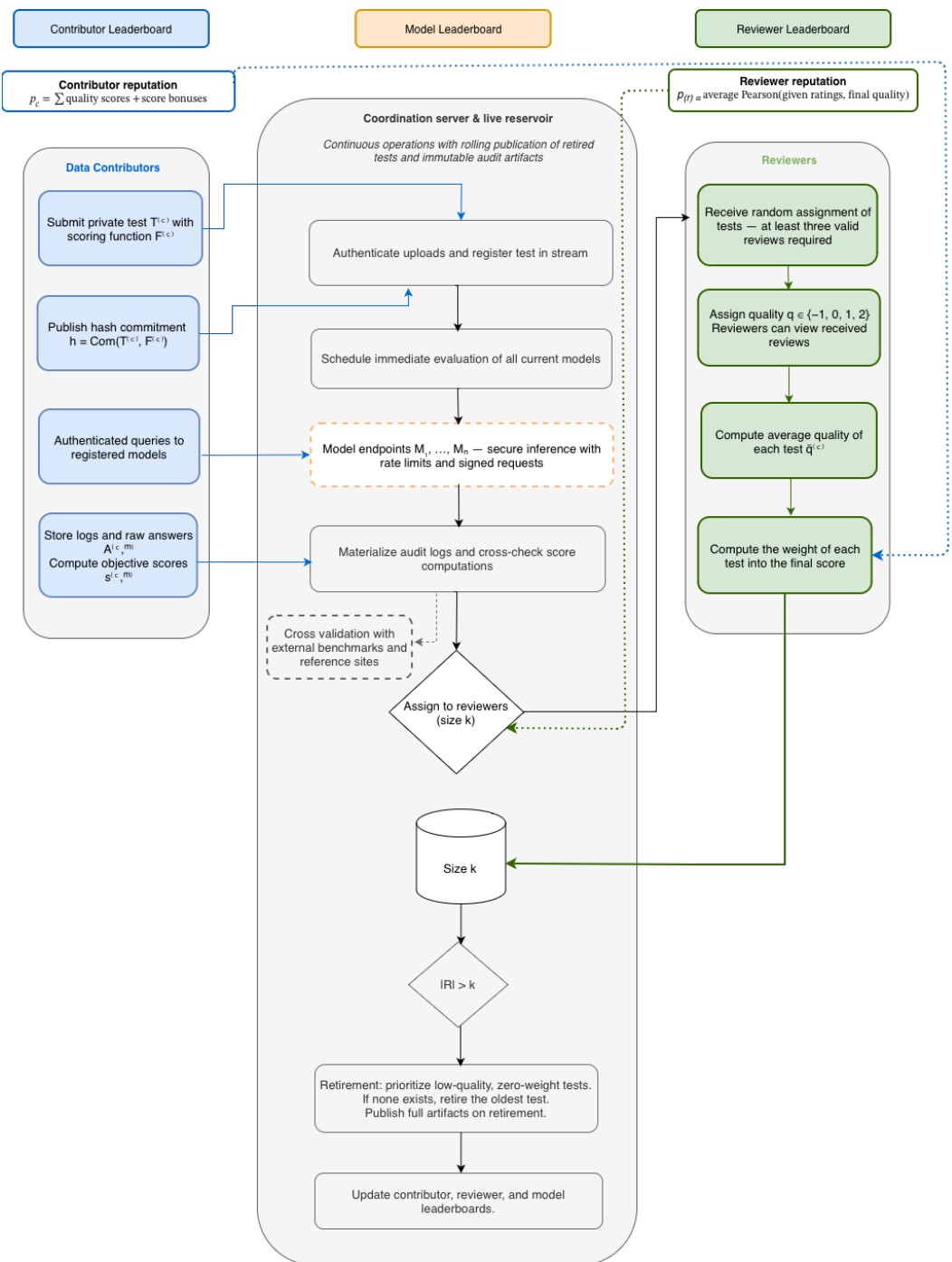

Figure 1: Overall architecture and continuous workflow in PEERBENCH. Blue denotes data contributors. Gray denotes the coordination service and the live reservoir. Green denotes reviewers. Orange denotes model endpoints. Dashed arrows show reputation feedback and cross validation links.

