# OpenReview forum: "Position: Benchmarking is Broken - Don't Let AI be Its Own Judge"
_NeurIPS.cc/2025/Position_Paper_Track — NeurIPS 2025 Position Paper Track_

### Official Review · Reviewer_4tZE · 2025-07-13

**Significance:** 4
**Presentation:** 3
**Rating:** 8
**Confidence:** 3

**Summary:**

This position paper argues that current AI benchmarking practices are fundamentally broken due to data contamination (test sets leaking into training data), selective reporting, systematic bias, fragmented metrics, and lack of quality control. These flaws make it virtually impossible to disentangle genuine progress from manufactured performance, undermining scientific credibility.

The paper advocates replacing current "open-book" self-reported leaderboards with standardized, proctored examinations similar to human examinations. Its solution centers on secret test sets, proctored execution, community governance, continuous renewal, cryptographic auditability, equitable access, and multi-metric reporting.

It presents PEERBENCH, a prototype platform implementing these principles through validators who create private test suites, reputation-based quality control, cryptographic verification, and rolling retirement of materials. The system aims to be contamination-resistant and governed by distributed stakeholders rather than centralized authorities.

The paper's contribution lies in its structural critique of existing evaluation practices and concrete roadmap for establishing trustworthy, sustainable AI assessment methods.

**Strengths:**

The paper presents a clear argument that current AI benchmarking practices are fundamentally broken and require systematic reform. The authors articulate their position with precision, advocating for a shift from "open-book" self-reported leaderboards to standardized, proctored examinations modeled after human tests.

The argument is well-supported with concrete evidence including specific contamination rates, documented cases of selective reporting, and systematic analysis of current failure modes. The authors provide theoretical reasoning about why contamination is inevitable with internet-scale training and practical examples of how benchmark gaming occurs.

The topic's NeurIPS relevance is high. Reliable evaluation underpins much machine learning research. The paper addresses an important problem of scientific credibility: benchmark scores may no longer reflect genuine capabilities. The PEERBENCH prototype demonstrates technical feasibility.

The authors thoughtfully address counterarguments in Section 5, showing awareness of legitimate concerns about their proposal while providing substantive responses.

**Weaknesses:**

The paper could benefit from empirical validation of PEERBENCH. The game-theoretic analysis of validator incentives needs deeper examination, as the reputation system could create new gaming opportunities. Cost-benefit analysis is superficial, lacking concrete estimates for infrastructure, human resources, and participation barriers that might exclude smaller research groups.

Alternative positions not adequately addressed include competition naturally eliminating unreliable benchmarks, regulatory standardization similar to clinical trials, industry self-regulation through research consortiums, and incremental reform focused on improving contamination detection. The paper also doesn't address whether some degree of benchmark gaming might have benefits, such as driving innovation, and it doesn't consider evaluations beyond exam-style ones, such as continuous real-world deployment metrics or collaborative/decentralized evaluation.

The subtitle does not accurately reflect the paper's position. Occasional capitalization/formatting inconsistencies appear unprofessional.

**Questions:**

How would PEERBENCH be economically sustainable? Who bears the costs? Would submission fees exclude smaller research groups or academic institutions with limited funding? This could influence its feasibility.

What evidence supports the claim that a complete paradigm shift is necessary instead of incremental reforms like better contamination detection, mandatory pre-registration of evaluation methods, or industry standards for benchmark retirement?

How would PEERBENCH prevent new forms of gaming, such as validator coalitions manipulating reputation scores, strategic timing of submissions, or validators reverse-engineering patterns from revealed test subsets? How would it manage the existing problem of sycophantic gaming?

Adoption requires mass coordination across competing organizations. Why is this feasible? What is your transition strategy?

For subjective tasks requiring human judgment, how would peer voting among validators avoid the biases you criticize in current LLM judging systems?

What would constitute failure conditions for PEERBENCH that would warrant returning to current approaches, and how would you detect such failures before they undermine scientific progress?

**Alternative Position:**

Yes, and alternative positions are well-considered and addressed by the argument

**Author Identification:**

No.

**Context:**

3

**Discussion:**

4

**Ethics:**

["NO or VERY MINOR ethics concerns only"]

**Position:**

Yes, the paper argues for or against a position related to machine learning.

**Support:**

3

**Thoroughness:**

2

---

### Official Review · Reviewer_yP8E · 2025-07-24

**Significance:** 2
**Presentation:** 3
**Rating:** 4
**Confidence:** 3

**Summary:**

The paper critiques current AI benchmarking practices, highlighting critical limitations such as data contamination, lack of quality control, selective reporting, and fragmented evaluation standards. To address these issues, the authors propose PeerBench—a community-governed evaluation framework inspired by standardized human testing systems like the SAT or GRE. The proposed system incorporates features such as secret test sets, cryptographically verified execution environments, a reputation-based validator network, and continuous test renewal, aiming to create a contamination-resistant yet research-useful benchmarking ecosystem.

**Strengths:**

1. The paper addresses an urgent and widely recognized issue in the AI community - the growing unreliability of existing benchmarks as indicators of real-world model capability.
2. The authors provide compelling examples and analysis of systemic benchmarking failures, including training data leakage and selective reporting, which support the motivation for their proposal.
3. The concept of PeerBench introduces novel elements such as cryptographic verification and continuous test renewal, which could offer meaningful improvements in benchmark robustness and credibility.

**Weaknesses:**

1. Scalability is a major concern of the proposed approach. The system depends on sustained human participation for test development and validation.
2. The described framework appears more accessible to well-funded organizations, which may inadvertently reinforce existing disparities in the research ecosystem.
3. The proposal currently lacks empirical validation—there are no experiments or prototype implementations to assess feasibility or impact.

**Questions:**

1. It would be helpful for the authors to clarify their position on the concerns raised above, especially with regard to scalability and inclusiveness.
2. The paper reads more like a proposal for a new benchmark than a broader position paper. It would benefit from clearly distinguishing the overarching position from the specific PeerBench design, allowing space for alternative community-driven solutions to emerge.

**Alternative Position:**

Yes, and alternative positions are well-considered and addressed by the argument

**Author Identification:**

No.

**Context:**

3

**Discussion:**

3

**Ethics:**

["NO or VERY MINOR ethics concerns only"]

**Position:**

Yes, the paper argues for or against a position related to machine learning.

**Support:**

2

**Thoroughness:**

4

---

### Official Review · Reviewer_TN1k · 2025-08-06

**Significance:** 3
**Presentation:** 3
**Rating:** 6
**Confidence:** 4

**Summary:**

The paper argues that current AI benchmarking practices are fundamentally flawed due to issues like data contamination, selective reporting, biased test construction, and lack of transparency. The paper calls for a shift toward robust, trustworthy, and auditable benchmarks to ensure that AI progress is accurately and fairly measured. To this end, the authors propose a new evaluation system, PEERBENCH, a prototype platform designed to implement this vision, using reputation-weighted validation, rolling test updates, and cryptographic auditability.

**Strengths:**

1. Clear: The paper clearly articulates a timely and pressing position: that current AI benchmarking practices are inadequate, prone to contamination, and increasingly untrustworthy.
2. Analytical: The authors support their position with a comprehensive critique of structural flaws in existing benchmarks, like  cherry-picking, selective reporting, lack of fairness, and data contamination.
3. Strong proposal: PEERBENCH is a strong prototype proposal that illustrates the feasibility of the vision, grounding the argument in practical design.
4. Counter-arguments: the paper engages with counterarguments and transparently acknowledges trade-offs, which strengthens its credibility.

**Weaknesses:**

1. While the paper presents a compelling case, it could more critically examine the limitations of using human standardized exams as the guiding metaphor for AI evaluation.
2. While the paper briefly mentions infrastructure and participation challenges, the scalability, economic viability, and governance logistics of PEERBENCH deserve deeper exploration.
3. The paper could also benefit from engaging more substantively with alternative proposals. For example, consider hybrid models that combine open and closed testing, or community-driven, dynamic benchmark curation.

**Questions:**

1. How does PEERBENCH plan to manage the scalability and sustainability of continuous test generation and validator participation over time, especially as the number of models and evaluation streams continue to grow?
2. Human exams are known to suffer from biases and systemic inequities. Why is this paradigm considered a gold standard for AI evaluation, and how does PEERBENCH avoid inheriting those flaws?
3. Could you elaborate on the tradeoffs between transparency and secrecy in evaluation? How does the system ensure community trust if critical test sets are withheld until retirement?
4. How would PEERBENCH handle adversarial or low-quality validator participation at scale, and what are the fallback mechanisms if validator consensus becomes fragmented or manipulated?
5. Have you considered or experimented with hybrid benchmarking models that incorporate both real-time user feedback and proctored-style evaluations?

**Alternative Position:**

Yes, and alternative positions are well-considered and named but not addressed

**Author Identification:**

No.

**Context:**

2

**Details Of Ethics Concerns:**

No major concerns

**Discussion:**

3

**Ethics:**

["NO or VERY MINOR ethics concerns only"]

**Position:**

Yes, the paper argues for or against a position related to machine learning.

**Support:**

3

**Thoroughness:**

4

---

### Note · Authors · 2025-08-27

**1-10 Additional Comments:**

We are very pleased with our experience and found the reviews thoughtful and helpful in sharpening our argument. The position paper track serves a vital function, and we strongly support its continuation.

A key point of feedback for the track itself would be to continue clarifying the evaluation criteria unique to position papers. Because the goal is to argue a position and spark discussion, many of the weaknesses cited in our reviews (especially for Review TN1k and yP8E in our submission), such as the lack of empirical validation or a fully specified implementation, do not hold in the same way they would for a technical paper. These are not flaws in the argument, but rather the very starting points for the future community work the paper aims to inspire.

Emphasizing this distinction in reviewer guidelines would help all participants align on the track's goals and reinforce its unique value to the community.

**1-11 Submit Again:**

Probably yes

**1-1 Submission Process:**

4

**1-2 Next Year:**

I would be very supportive of continuing the position paper track. It provides a valuable forum for meta-level discussions that are crucial for the health and direction of the field. For next year, it might be beneficial to organize dedicated sessions or panels where authors of accepted position papers can engage in a structured debate or discussion on their respective topics. This could amplify the impact of the positions and foster a more direct community dialogue, which is the ultimate goal of this track.

**1-3 Future Development:**

To build on the previous point, creating a more integrated experience beyond publication could be powerful. For instance, pairing papers with opposing viewpoints for a joint presentation or a 'Great Debates' style session could be highly engaging. Additionally, a mentorship program connecting junior researchers with senior figures to develop bold positions could encourage more ambitious and diverse submissions.

To further enhance the review process and mitigate potential misunderstandings between authors and reviewers, future iterations of the track might benefit from more detailed guidelines. Providing clearer expectations on the appropriate balance between high-level conceptual arguments and technical specifics would be valuable. Additionally, offering criteria that distinguish a rigorous academic position paper from an informal blog post, particularly regarding novelty, structured evidence, and engagement with existing literature, could help ensure all participants share a common understanding of the track's standards and goals.

**1-4 Interest:**

["Panel discussions with other position paper authors", "Structured debates on controversial topics", "Workshops for developing position papers", "Mentorship programs for early-career researchers"]

**1-5 Thoughtful:**

9

**1-6 Supportive:**

7

**1-7 Technical Aspects Versus Position:**

6

**1-8 Gate Keeping:**

8

**1-9 Camera Ready Changes:**

We are very grateful for the constructive feedback from all reviewers. They will significantly improve the clarity and impact of our paper. The planned changes for the camera-ready version are centered on sharpening our core message and clarifying the scope of our contribution. Our primary revisions will be:

- Sharpening our Core Message: The most important change, prompted by the reviews, will be to restructure the narrative to more explicitly lead with our core position: that the current AI evaluation paradigm is structurally broken and requires a fundamental shift. We will elevate our proposed principles for trustworthy evaluation to make it clear they are the central contribution.

- Clarifying the Role of PeerBench: We will explicitly frame PeerBench as a conceptual prototype and an existence proof. Its purpose is not to be a final, fully-specified system, but to demonstrate that our principles are feasible and to serve as a concrete starting point for a community-wide discussion. This will ensure that the paper isn't yet another a benchmark proposal, but a position paper.

- Expanding Discussion on Alternatives & Challenges: We will strengthen our engagement with alternative solutions, such as market forces, regulatory bodies, and industry consortia, to better contextualize our argument. We will also more explicitly identify the practical challenges (scalability, equity, potential for gaming) associated with our proposed direction, framing them as the critical research agenda our paper seeks to inspire.

- Refining the Argument in Response to Specific Questions: We will integrate our responses to the reviewers' excellent questions to clarify key points. These are detailed in our responses to specific reviewers in Section 3.

In essence, our camera-ready version will ensure the final paper is laser-focused on its primary goal: to persuade the community of the urgency of evaluation reform and to catalyze a collective effort to build a better path forward.

**3-1 Review Response1:**

TN1k

**3-2 Reaction To Review1:**

We thank the reviewer for the insightful feedback and for affirming our core position. The excellent questions raised mostly target the practical implementation of PeerBench, our conceptual prototype. As the reviewer correctly implies, these challenges do not undermine our paper's position; rather, they highlight the critical discussion we aim to ignite, and is precisely the conversation we hope to start.

In response to the questions:

- Scalability and sustainability: We agree these are critical challenges. Our proposed reward systems are designed specifically to incentivize sustainable, high-quality human participation in PeerBench. However, we will also clarify in the revision that PeerBench is intended as a conceptual prototype model to spark discussion; a full implementation and wide use would require significant community investment.

- Human Exam Metaphor: Our intention was not to hold them up as a gold standard free of bias. Rather, we use the analogy to argue that AI evaluation deserves at least the level of seriousness, proctoring, and structured design applied to high-stakes human tests. We will refine our language in the revision to make it clear that we aim to learn from, not perfectly replicate, this paradigm, while actively designing mechanisms to mitigate biases in human testing.

- Transparency vs Secrecy: Our position is that temporary secrecy is a pragmatic tool to ensure the integrity of the evaluation and prevent data contamination. Long-term trust is built through transparency, auditability and the eventual public release of retired test sets, ensuring the entire process is accountable in hindsight.

- Adversaries: Reputation system is the primary defense, which algorithmically penalize malicious contributions. We acknowledge that no community-based ecosystem can withstand a majority of bad actors, and thus we operate under a standard assumption that most participants are honest, and the system's incentives will marginalize those who do not.

**3-3 Review Response2:**

yP8E

**3-4 Reaction To Review2:**

We thank the reviewer for the critical feedback, which is invaluable for clarifying our paper's scope.

The concern that the paper reads more like a benchmark proposal than a position paper, is a helpful insight. Our position is that the current evaluation paradigm is critically flawed and broken. PeerBench is offered only as a concrete prototype to demonstrate that this issue we propose is actually solvable, which strengthens our position. We will sharpen this distinction in our revision by explicitly framing PeerBench as a conceptual model intended to strengthen our argument and spark discussion on possible alternatives.

The other concerns regarding scalability, equity, and experiments target the prototype implementation. These don't hinder our paper's core position, rather, they highlight the exact challenges we believe the community should discuss and address together.

- On Scalability & Equity: We propose mechanisms like reputation-weighting and reward systems in the conceptual model. Our framework is designed to promote equity by shifting from the current resource-driven paradigm to one based on merit and fairness. We believe that, beyond our mechanism design, a truly robust solution also requires a community-wide effort and discussion, and this is what our paper aims to catalyze.

- On Lack of Experiments: This is intentional. Our goal is not to prematurely validate one implementation, but to first build consensus that a new paradigm is needed. Our roadmap is
(1) Use this paper to build consensus that our current evaluation paradigm is critically flawed.
(2) Discuss potential solutions, treating PeerBench as just one candidate prototype to spark debate.
(3) Collectively refine the best path forward. Empirical validation is the crucial final step on that shared path.
Ultimately, our aim is to bring the critical issue to the community's attention and harness our collective effort to build a more trustworthy evaluation ecosystem that benefits everyone.

**3-5 Review Response3:**

4tZE

**3-6 Reaction To Review3:**

We are extremely grateful to the reviewer for the strong support and exceptionally insightful feedback. The analysis affirms our core position and provides an excellent roadmap for future discussion.

The reviewer raises an excellent point about more thoroughly addressing alternative solutions, which we will clarify in our revision:

- Regarding market competition, our position is that market forces are not just slow, but can be actively misleading. In practice, contaminated, saturated, or biased benchmarks (e.g. the case analysis of GLUE, SUPERGLUE, Humanity's Last Exam that we identify in the paper) have remained dominant long after their scientific signal has decayed, driven by distorted incentives. Relying solely on the market to self-correct is a reactive approach; it's like knowing a scam is active but waiting for people to be defrauded before acting. We believe a proactive, structural intervention is required and preferable when possible.

- Regarding regulatory or consortium approaches, we acknowledge their value but caution against over-centralization. As discussed in our paper, proprietary or privately controlled benchmarks create significant risks of capture and opacity. Our proposed approach differs by design: Its governance is decentralized, validator-weighted, and auditable, reducing reliance on any single entity while retaining accountability.

The reviewer also poses critical questions about PeerBench's path to adoption. We agree entirely. These are precisely the necessary, difficult questions that we hope our paper encourages the entire community to engage with. Our goal with this position paper is to highlight the urgency of the problem and open up the non-trivial challenges as avenues for future work. We will explicitly frame them in our conclusion as an open invitation and call to action for community-wide research and development.

We are deeply grateful for the highly constructive and encouraging review; It is invaluable in refining our paper.

---

### Meta-Review · Area_Chair_ryKp · 2025-08-20

**Rating:** 7
**Confidence:** 4

**Strengths:**

The paper takes a clear and timely stance on the inadequacy of current AI benchmarks, offering a rigorous critique supported by compelling examples of systemic failures. The novel PeerBench proposal demonstrates feasibility through a practical prototype and introduces innovations like cryptographic verification and continuous renewal. By acknowledging trade-offs and counterarguments, the paper maintains credibility while addressing a pressing issue for the AI community.

**Weaknesses:**

A relatively more major concern is that scalability, economic viability, and governance logistics remain underexplored, especially given their reliance on human participation. This paper also does not address alternative approaches and does not consider evaluations beyond exam-style ones. Two reviewers are concerned about lacking empirical validation, offering no prototype, experiments, or concrete cost–benefit analysis.

**Questions:**

1.	How does PEERBENCH plan to manage the scalability and sustainability of continuous test generation and validator participation over time, especially as the number of models and evaluation streams continue to grow?
2.	Why is this paradigm considered human exam as a gold standard for AI evaluation, and how does PEERBENCH avoid inheriting those flaws?
3.	Could you elaborate on the tradeoffs between transparency and secrecy in evaluation? How does the system ensure community trust if critical test sets are withheld until retirement?
4.	How would PEERBENCH handle adversarial or low-quality validator participation at scale, and what are the fallback mechanisms if validator consensus becomes fragmented or manipulated?
5.	Have you considered or experimented with hybrid benchmarking models that incorporate both real-time user feedback and proctored-style evaluations?
6.	How would PEERBENCH prevent new forms of gaming, such as validator coalitions manipulating reputation scores, strategic timing of submissions, or validators reverse-engineering patterns from revealed test subsets?
7.	What would constitute failure conditions for PEERBENCH that would warrant returning to current approaches?

**Ethics:**

No.

**Thoroughness:**

3

---

### Decision · Program_Chairs · 2025-09-26

Accept